# Are Preoperative CT Findings Useful in Predicting the Duration of Laparoscopic Appendectomy in Pediatric Patients? A Single Center Study

**DOI:** 10.3390/jcm13185504

**Published:** 2024-09-18

**Authors:** Ismail Taskent, Bunyamin Ece, Mehmet Ali Narsat

**Affiliations:** 1Department of Radiology, Kastamonu University, 37150 Kastamonu, Turkey; bunyaminece@kastamonu.edu.tr; 2Department of Pediatric Surgery, Kastamonu University, 37150 Kastamonu, Turkey; manarsat@kastamonu.edu.tr

**Keywords:** acute appendicitis, pediatric, CT findings, laparoscopic surgery, operation time, predictors

## Abstract

**Background/Objectives:** Preoperative computed tomography (CT) imaging plays a vital role in accurately diagnosing acute appendicitis and assessing the severity of the condition, as well as the complexity of the surgical procedure. CT imaging provides detailed information on the anatomical and pathological aspects of appendicitis, allowing surgeons to anticipate technical challenges and select the most appropriate surgical approach. This retrospective study aimed to investigate the correlation between preoperative CT findings and the duration of laparoscopic appendectomy (LA) in pediatric patients. **Methods:** This retrospective study included 104 pediatric patients diagnosed with acute appendicitis via contrast-enhanced CT who subsequently underwent laparoscopic appendectomy (LA) between November 2021 and February 2024. CT images were meticulously reviewed by two experienced radiologists blinded to the clinical and surgical outcomes. The severity of appendicitis was evaluated using a five-point scale based on the presence of periappendiceal fat, fluid, extraluminal air, and abscesses. **Results:** The average operation time was 51.1 ± 21.6 min. Correlation analysis revealed significant positive associations between operation time and neutrophil count (*p* = 0.014), C-reactive protein levels (*p* = 0.002), symptom-to-operation time (*p* = 0.004), and appendix diameter (*p* = 0.017). The total CT score also showed a significant correlation with operation time (*p* < 0.001). Multiple regression analysis demonstrated that a symptom duration of more than 2 days (*p* = 0.047), time from CT to surgery (*p* = 0.039), and the presence of a periappendiceal abscess (*p* = 0.005) were independent predictors of prolonged operation time. In the perforated appendicitis group, the presence of a periappendiceal abscess on CT was significantly associated with prolonged operation time (*p* = 0.020). In the non-perforated group, the presence of periappendiceal fluid was significantly related to longer operation times (*p* = 0.026). **Conclusions:** In our study, preoperative CT findings, particularly the presence of a periappendiceal abscess, were significantly associated with prolonged operation times in pediatric patients undergoing laparoscopic appendectomy. Elevated CRP levels, the time between CT imaging and surgery, and a symptom duration of more than 2 days were also found to significantly impact the procedure’s duration.

## 1. Introduction

Acute appendicitis, a common cause of abdominal pain in children, often necessitates swift surgical intervention for accurate diagnosis [1]. The utilization of preoperative computed tomography (CT) imaging has notably enhanced diagnostic precision, particularly in cases where clinical and laboratory findings are inconclusive, thus minimizing unnecessary surgeries [2,3]. CT plays a pivotal role in accurately diagnosing acute appendicitis and holds the capability to predict histological severity. This capability facilitates informed decision-making regarding the timing and necessity of operative intervention [4].

Either laparoscopic or open surgery is the preferred surgical approach to treat pediatric patients with acute appendicitis. However, laparoscopic appendectomy (LA) holds several advantages over the open approach. These include shorter recovery times, reduced postoperative pain, a faster return to normal activities, and a more favorable cosmetic outcome [5,6,7,8,9]. However, operation times have been observed to be longer in patients who underwent LA compared to those who had open appendectomy (OA) [5,10].

Operation time is a reliable measure of technical difficulty in laparoscopic procedures, and research shows a link between prolonged operation times and increased complication rates [11,12,13]. Imaging findings, particularly CT scans, have an important role in predicting surgery difficulty and outcomes. CT can easily detect the presence of periappendiceal fluid, extraluminal air, and abscesses, as well as the diameter and position of the appendix, which are crucial in determining the surgical difficulty. These imaging results may enhance surgical outcomes by allowing surgeons to anticipate difficulties and plan accordingly [14,15]. To the best of our knowledge, there are not enough studies demonstrating the relationship between prolonged operation time and CT findings in LA in the pediatric population.

The purpose of this retrospective study is to investigate the relationship between preoperative CT imaging findings and operation time for LA in pediatric patients.

## 2. Materials and Methods

### 2.1. Participants

Pediatric patients who presented to our hospital with acute abdomen symptoms and were suspected of having acute appendicitis but could not be diagnosed with ultrasound, requiring a contrast-enhanced CT, between November 2021 and February 2024, were retrospectively evaluated. All surgeries were performed by a single pediatric surgery specialist at our hospital. During this period, a total of 138 patients were diagnosed with appendicitis via CT, and 127 of them underwent surgical intervention, while the remaining patients were treated with non-surgical methods such as drainage and antibiotics. Out of 386 total appendectomy patients, 127 underwent preoperative CT evaluation. The remaining patients were evaluated using ultrasound imaging. Patients who underwent non-contrast CT (*n* = 8), those whose CT records were inaccessible (*n* = 9), and those with incomplete medical records (*n* = 4) were excluded from this study. Additionally, 2 patients with significant comorbidities that could have affected the operation time were excluded from this study. As a result, a total of 104 patients diagnosed with acute appendicitis on contrast-enhanced CT and underwent LA were included in this study.

Ethical approval was obtained from the institutional review board of our tertiary care center (ethics committee number: KAEK 2022-104, decision date: 19 October 2022) for the retrospective review of laboratory findings and CT scans in patients who underwent LA. This study adhered to the “Declaration of Helsinki”. Because this study was retrospective, informed consent was not obtained.

### 2.2. Data Collection

The data were obtained from the surgical records, anesthesia records, and progress notes from the pediatric surgery clinic at our hospital. Perforated appendicitis was defined as the spillage of appendiceal contents, peritonitis, or abscesses observed at the beginning of surgery. Operation time was defined as the duration from the initiation of the skin incision to the completion of skin closure, based on anesthesia records. Patients were divided into two groups according to operation time: those with operation times ≤ 50 min and those with operation times > 50 min. Preoperative parameters included in the analysis were age, gender, body mass index (BMI), white blood cell (WBC) count, neutrophil count (NEU), NEU/WBC ratio, C-reactive protein (CRP) level (normal range in our lab: 0–5 mg/L), heart rate, body temperature, symptom duration (symptom-to-operation time), waiting time (CT to operation time), and findings from CT. To assess the relationship between operation time and outcomes, the complications, readmissions, and length of hospital stay (LOS) for both groups were compared.

### 2.3. Abdominal CT Assessment

All CT investigations were conducted using 64-detector, 128-slice CT scanners (Revolution EVO; GE Medical Systems, Chicago, IL, USA). All examinations were performed with a low-dose technique utilizing an automated tube current modulation to determine the tube current (mA). All CT scans were conducted at 80–120 kV, with adjustments made automatically based on the patient’s physique. The standard section slice thickness was 0.625 mm. The contrast material used (300 mg iodine/mL) amounted to 1 mL/kg. The contrast material was administered automatically. CT examinations were consistently conducted in the portal venous phase.

To obtain a consensus, two radiologists (I.T. with 11 years and B.E. with 10 years of experience) assessed the patients’ CT scans using an Advantage Windows workstation (ADW 4.7 Ext. 16 Software, GE Medical Systems, Chicago, IL, USA). The radiologists were not aware of the clinical or surgical results (Figure 1).

To classify the localizations, three regions were defined in the appendix. An anterior location (where the end of appendix lies anterior to cecum in the large pelvis) was labeled as “Location 1”, a retrocecal or retrocolonic location (where the tip of appendix lies behind the cecum, in the right iliac fossa or reaches the subhepatic area) was labeled as “Location 2”, and a pelvic location (where the tip of the appendix lies in the small pelvis) was labeled as “Location 3” [16].

Appendicitis findings were graded based on imaging features observed in the periappendix region. A 5-point scale was devised, considering the presence of an appendicolith, periappendiceal fat stranding, periappendiceal fluid, extraluminal air, and abscess finding. The presence of any of these findings contributed 1 point to the rating, which ranged from 0 to 5 [17].

### 2.4. Surgical Procedure

All surgeries were performed by the same surgeon with the patient under general anesthesia. A 10 mm optic port was inserted into the umbilicus or subumbilical position, and an additional 5 mm working port was inserted into the left iliac fossa position. A needle grasper was introduced into the abdomen from the right iliac fossa, or a suprapubic or right subcostal 5 mm trocar was used, depending on the location and clinical appearance of the appendix during the initial exploration.

Once a pneumoperitoneum pressure of 8–15 mm Hg was established using CO_2_ at a flow rate of 3–5 L per minute, the patient was positioned in a 30° Trendelenburg position with a 15° left tilt. The mesoappendix was dissected using laparoscopic coagulating shears, and the base of the appendix was ligated with endo-clips. After the appendix was divided above the ligated site using laparoscopic coagulating shears, the resected appendix was extracted through the transumbilical port. Subsequently, drainage catheters were inserted through the suprapubic port site into the pelvic cavity or paracolic gutter, depending on the operative findings related to peritoneal contamination from appendiceal perforation and the surgeon’s judgment.

No patients required blood transfusion during the perioperative period. For postoperative antibiotic prophylaxis, second-generation cephalosporin and metronidazole were administered.

### 2.5. Statistical Analysis

The data were analyzed using the Statistical Package for the Social Sciences (SPSS) for Windows version 23 software (IBM SPSS Inc., Chicago, IL, USA). Normal distribution of the data was assessed using the Kolmogorov–Smirnov test. Numerical variables with a normal distribution are presented as mean ± standard deviation (SD) values, while variables without a normal distribution are presented as median (minimum–maximum) values. Categorical variables are reported as the number (*n*) and percentage (%). The Chi-square test was employed to compare categorical variables. For group comparisons, the independent samples *t*-test was used for data with a normal distribution, and the Mann–Whitney U test was used for data without a normal distribution. Pearson correlation analysis was utilized for data with a normal distribution, and Spearman correlation analysis was applied for data without a normal distribution. In univariate analysis, simple linear regression was used to evaluate the relationship between the operation time and each independent variable. A multivariable linear regression analysis was performed to assess the impact of various independent variables on operation time. A significance level of *p* < 0.05 was considered statistically significant.

## 3. Results

This study included 59 (56.7%) males and 45 (43.3%) females, with an average age of 11.7 ± 4.1 years (range: 2.8 to 18 years). The average surgery duration was 51.1 ± 21.6 min. The average time from symptom onset to operation was 32.8 ± 33 h, while from CT scan to operation was 9.4 ± 10.3 h. Appendicoliths were discovered in 55 (52.9%) instances, periappendiceal fat stranding in 86 (82.7%) cases, extraluminal air in 21 (20.2%), ascites in 42 (40.4%) cases, and symptoms of abscess in 15 (14.4%) patients. The average appendix diameter was 11.0 ± 3.2 mm, with females measuring 10.4 ± 3.0 mm and males measuring 11.5 ± 3.2 mm. The appendix vermiformis was classified as follows: just below the anterior abdominal wall (Location 1) in 39 (37.5%) patients, in the retrocecal or retro-ascending colon (Location 2) in 26 (25.0%) patients, and in the pelvis covered by the small intestine (Location 3) in 39 (37.5%) patients. Postoperative complications included wound infections in seven (6.7%) patients and a single case (1.0%) of postoperative abscess. Three patients (2.9%) required readmission. The mean length of stay (LOS) in the hospital was 3.75 ± 1 days, and the mean body temperature was 37.61 ± 0.5 °C. The average heart rate was recorded at 90.42 ± 9.73 beats per minute (BPM) (Table 1).

The study population had an average age of 140.9 ± 49.3 months. The average BMI was 20.7 ± 4.4. The WBC and NEU counts were 15.2 ± 5.3 and 12.2 ± 5.3 µL, respectively, indicating usual inflammatory responses. Participants had different amounts of inflammation, with an average CRP level of 68.3 ± 78.3 mg/L. The average time between symptom to operation is 32.8 ± 33.0 h, and the average time from CT to operation is 9.5 ± 10.3. The comparison of demographic and clinical data based on gender indicates that no statistically significant differences were observed (*p* > 0.05) (Table 2).

According to the correlation analysis results presented in Table 3, the relationships between operation time and continuous variables were evaluated. Analysis results revealed that some variables showed statistically significant effects on operation time. NEU (*p* = 0.014), CRP levels (*p* = 0.002), time from symptom onset to operation (*p* = 0.004), and appendicitis diameter (*p* = 0.017) were positively significant between operation time. On the other hand, other variables such as age, height, weight, BMI, WBC, BPM, temperature, and time from CT to operation did not show a significant relationship with operation time (*p* > 0.05) (Table 3).

The total score, ranging from zero to five, showed a significant and positive correlation with operation time in the correlation analysis (correlation coefficient = 0.415, *p* < 0.001, Figure 2).

In the analysis, to evaluate the effects of removing specific components from the model on operation time, the results showed that the inclusion or exclusion of certain variables significantly impacted model performance. When all components were included, the model showed an R^2^ of 0.359, indicating that 35.9% of the variance in operation time was explained by the predictors, with an F-value of 11.001 (*p* < 0.001). Removing the abscess variable resulted in a drop in R^2^ to 0.271 and an increased standard error of the estimate (SEE) to 18.883. Further removals of extraluminal air and periappendicular fluid continued to decrease the model’s explanatory power, with the removal of periappendicular fluid lowering R^2^ to 0.083 and the F-value to 4.575 (*p* = 0.013). Finally, removing periappendicular fat-stranding reduced R^2^ to 0.029, with the model’s performance becoming marginally significant (F = 3.012, *p* = 0.086). Based on the analysis, the abscess variable is the most important factor in the model, and its removal significantly decreases the model’s explanatory power and performance (Table 4).

The relationships between categorical and continuous variables with operation time were evaluated with univariate and multivariable analysis, and the results are presented in Table 4 and Table 5. The results revealed that the presence of periappendiceal fat stranding, periappendiceal fluid, extraluminal air, and an abscess had a substantial impact on operation time. However, no significant association was found between appendicolitis, appendix location with operation time. CRP levels, symptom-to-operation time, CT-to-operation time, and appendix diameter showed a significant correlation with operation time. However, other factors such as age, BMI, WBC, and NEU did not have a significant impact on operation time (Table 5 and Table 6).

Our analysis of the relationship between symptom duration and operation time revealed that symptom duration does not have a linear effect on operation time. The second-degree polynomial model and LOESS method demonstrated significant fluctuations in operation time as symptom duration increases, indicating a nonlinear relationship (Figure 3). Based on this, we divided the patients into two groups: those with a symptom duration of 2 days or less and those with more than 2 days.

According to the multiple regression analysis results presented in Table 6, the effects of various variables on operation time were examined. The results show that a symptom duration exceeding 2 days (*p* = 0.047) and the time from CT to operation (*p* = 0.039) had a significant effect on operative time. Additionally, the presence of an abscess (*p* = 0.005) also significantly affects the operation time. Other variables, age, appendix diameter, BMI, NEU/WBC, CRP, presence of appendicolith, periappendiceal fat stranding, periappendiceal fluid, extraluminal air, and appendix localization, did not have a significant effect on the operation time (*p* > 0.05). The overall explanatory ratio R^2^ value of the model was “0.433”, indicating that the independent variables have a certain effect on the operation time (Table 7).

The median operation time was determined to be 50 min based on our distribution analysis. Using this median as a threshold, operation times exceeding 50 min were classified as long operation time groups. We then compared the clinical and laboratory characteristics between the groups to identify factors influencing prolonged versus shorter operation times. In the long operation time group, CRP levels were significantly higher (*p* = 0.002), symptom-to-operation time was longer (*p* < 0.001), and the duration of symptoms exceeding 2 days was more common (*p* = 0.016). Additionally, periappendiceal fluid (*p* < 0.001), extraluminal air (*p* < 0.001), and the presence of an abscess (*p* < 0.001) were more frequently observed in the long operation time group. No significant differences were found between the groups for other variables such as gender, age, BMI, and appendix diameter (Table 8).

In our study, we also conducted an analysis to examine the factors affecting operation time between two groups classified based on intraoperative findings: perforated and non-perforated appendicitis. In patients with perforated appendicitis, the presence of a periappendiceal abscess on CT was found to be significantly associated with prolonged operative time (*p* = 0.020). However, no significant differences were observed in age, gender, BMI, CRP, or other clinical parameters. In the non-perforated group, the presence of periappendiceal fluid on CT was significantly associated with prolonged operation time (*p* = 0.026), while no significant differences were noted for other variables (Table 9 and Table 10).

## 4. Discussion

In our study, preoperative CT findings were shown to play an important role in predicting the duration of surgery in pediatric patients undergoing laparoscopic appendectomy. Specifically, CT findings such as periappendiceal fluid, extraluminal air, and the presence of a periappendiceal abscess were significantly associated with prolonged operation times. Additionally, elevated CRP levels, the waiting time between the CT imaging and the surgery time, and a symptom duration of more than 2 days were found to be significantly related to prolonged operation times. Furthermore, this study discovered a link between the CT findings and the scoring system developed from these findings and the operation time.

There are several studies with diverse findings on the factors influencing the duration of laparoscopic appendectomy. Siewert et al. [2] demonstrated that preoperative CT findings can be used to predict operative time and complications during laparoscopic appendectomy. Hosokawa et al. [18] discovered that an increase in intra-abdominal fat density and a retrocecal or retro-ascending colon-located appendix prolonged the operation time in pediatric patients. Kohga et al. [19] discovered that the presence of free air was an independent predictor of the development of an intraabdominal abscess. These findings suggest that preoperative CT is a useful tool for predicting postoperative complications and estimating operative time, particularly in complex cases. Our study showed that higher total scores represent more complex surgical cases, and this prolongs the operation time. This finding revealed that as the total score increases, the operation time also increases, and surgeons spend more time dealing with more complex situations. Therefore, evaluating these parameters in the preoperative period may help surgeons predict the duration and degree of difficulty of the operation. However, in our multivariable regression analysis, only periappendiceal abscess was independently associated with operation time. Therefore, while the scoring system can be used as a guide to estimate overall surgical complexity, the independent effect of each component on operation time is limited.

Factors such as being overweight, having high C-reactive protein (CRP) levels, experiencing symptoms for more than 3 days, having an appendix diameter greater than 10 mm, the presence of free air, and the presence of an abscess on CT are findings supporting complicated appendicitis and all independent predictors of prolonged laparoscopic surgery time [12,20]. In cases of complicated appendicitis, additional procedures such as intra-abdominal irrigation and drainage tube placement are often required, which may prolong surgery time [21,22]. In the study conducted by Jeon et al., it was demonstrated that CRP exhibited a significant correlation with operation time, while WBC and NEU showed no correlation [12]. In our study, a significant relationship was found between increasing CRP levels and prolonged operation time in both univariate and multivariable analyses. However, while high WBC and NEU levels showed a positive correlation with surgical times, no significant relationship was observed in multivariable analyses. Furthermore, our study found that the duration of symptoms, CT operation time, and the presence of a periappendiceal abscess were directly related to operative time, with regression analysis confirming these as independent factors.

Although laparoscopic surgery was once considered a relative contraindication in overweight patients, this perception has evolved with increasing surgical experience. Enjoji et al. conducted a study revealing a correlation between an increase in BMI value and the duration of surgery in adult patients undergoing three-port laparoscopic appendectomy [11]. Another study by Enochsson et al. found that being overweight, defined as having a BMI > 26.4, significantly prolonged the duration of surgery in open appendectomy, but this negative impact of being overweight was not observed in laparoscopic appendectomy [16]. Contrarily, Hosokawa et al. demonstrated in their studies that there is no relationship between BMI and the duration of surgery [18,23]. Similarly, in our current study, no correlation was identified between BMI and operation time.

In our study, we also analyzed the factors affecting operation times in both perforated and non-perforated patient groups. In the study by Jeon et al. [12], the appendiceal diameter was associated with prolonged operation time in non-perforated appendicitis, whereas periappendiceal abscess was identified as a more significant factor in perforated cases. Similarly, in our study, the presence of a periappendiceal abscess was the most important factor contributing to prolonged operative time in the perforated group. However, in the non-perforated group, periappendiceal fluid was identified as the main factor influencing operation time. Appendiceal diameter did not have a significant effect in either group.

In rare anatomical positions, laparoscopic appendectomy proves to be a preferable option over the open technique. This preference arises from the surgeon’s ability to strategically select trocars and determine their placement once the camera is introduced, and the appendix is properly positioned, contrary to some studies indicating a statistically longer mean operative time in retrocecal and subhepatic groups compared to other anatomical positions [16,18,23]. Based on our study findings, the location of the appendix is not associated with the duration of the operation.

The strengths of our study increase the reliability and accuracy of our results. At first, this study is advantageous due to its substantial sample size, which provides high statistical power and the ability to apply the findings to a larger sample. In addition, the patient cohort demonstrated homogeneity, which minimized variation and enhanced the statistical significance of the results for a specific population group. Furthermore, it is important to note that the surgical procedures were consistently performed by a single surgeon, using the same technique. This conscious approach was carried out to eliminate any potential variations between surgeons and ensure that any variation in surgical outcomes could not be related to differences in surgical expertise or methodology. Together, these strengths enhance the long-term reliability of the findings from our study.

This study has some limitations. First, because this study was retrospective, we were unable to assess additional factors that could influence operation time, considering the fact that all procedures were performed by the same surgeon using the same technique. Second, the utilization of CT scans for patient evaluation, with the resulting radiation exposure, is an important obstacle. Third, due to the limited sample size, the relationship between prolonged operation time and postoperative outcomes could not be adequately assessed. Future prospective research with larger patient groups will provide further understanding of this topic.

## 5. Conclusions

In conclusion, our investigations indicate that preoperative CT findings, elevated CRP levels, and a symptom duration of more than 2 days have a significant impact on the duration of laparoscopic appendectomy. In particular, among the CT findings, the presence of a periappendiceal abscess emerged as the strongest independent predictor of operation time. While the five-point CT-based scoring system provides a general framework for assessing surgical complexity, its predictive power for operation time is limited, as not all components significantly influence surgical duration. Future studies with larger patient cohorts and more comprehensive analyses are needed to refine and validate these predictive models.

## Figures and Tables

**Figure 1 jcm-13-05504-f001:**
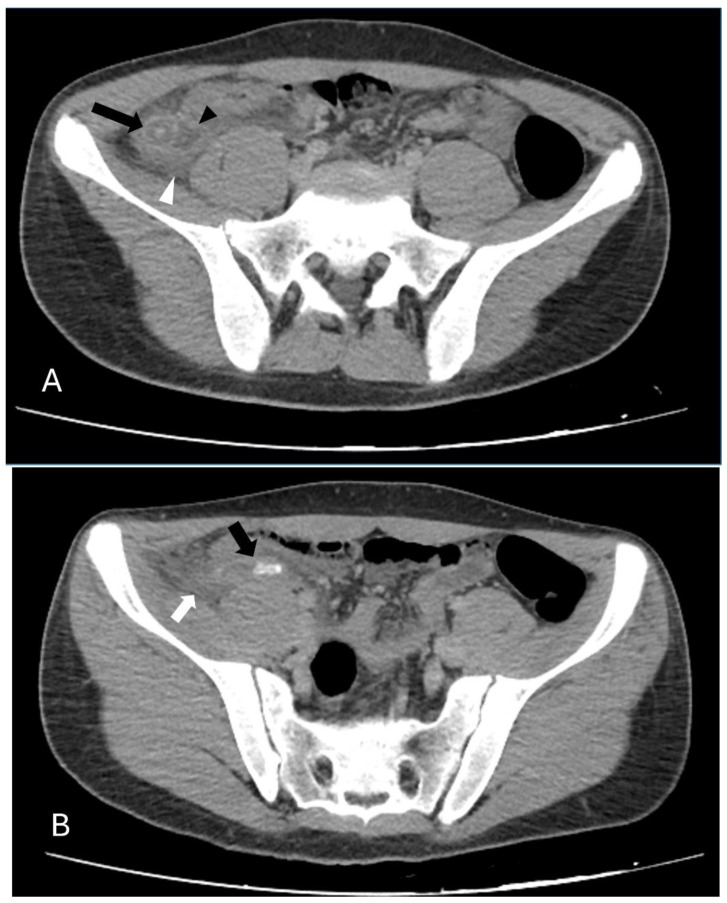
Axial CT images of a 189-month-old male patient. (**A**) The appendix (black arrow) exhibits an increased diameter and a thickened wall, indicative of inflammation. There is a notable collection of periappendiceal fluid (black arrowhead), and the surrounding fat tissue shows increased attenuation (white arrowhead), suggesting inflammatory changes. (**B**) In another section, an appendicolith (black arrow) is clearly visible. Additionally, there is increased attenuation in the periappendiceal fat tissue (white arrow), consistent with inflammatory changes associated with acute appendicitis.

**Figure 2 jcm-13-05504-f002:**
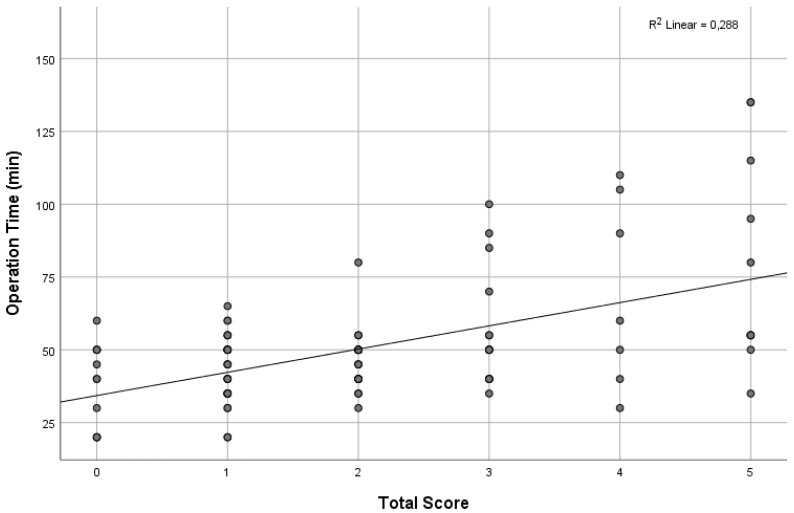
The impact of a scoring system comprising five variables (0 to 5) from CT findings on average operation time (min).

**Figure 3 jcm-13-05504-f003:**
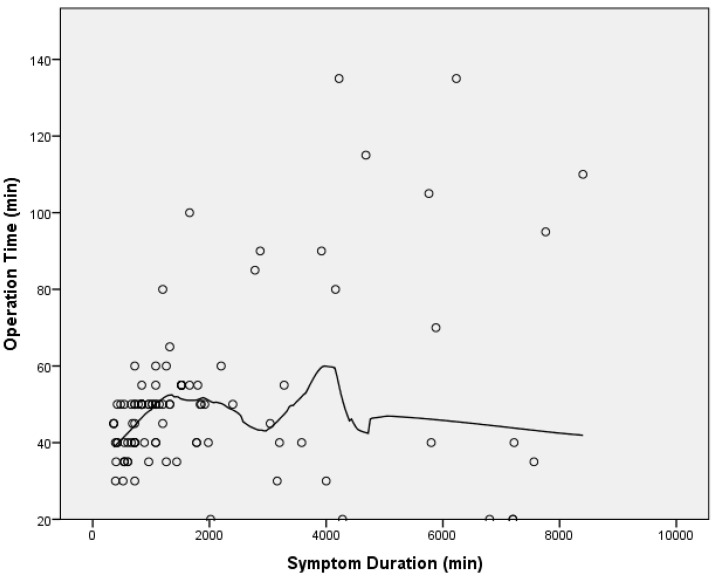
LOESS curve depicting the nonlinear relationship between symptom duration and operation time.

**Table 1 jcm-13-05504-t001:** The demographic and clinical data of the patients.

Variables	Value
Number of patients, *n*	104
Sex, *n*, (M/F)	59/45
Age (month)	140.9 ± 49.3
BMI (kg/m^2^)	20.7 ± 4.4
WBC (per µL)	15.2 ± 5.3
NEU (per µL)	12.2 ± 5.3
CRP (mg/L)	68.3 ± 78.3
Symptom–operation time (h)	32.8 ± 33.0
CT–operation time (h)	9.4 ± 10.3
Operation time (min)	51.1 ± 21.6
Appendix diameter (mm)	11.0 ± 3.0
BPM	90.4 ± 9.7
Temperature (°C)	37.6 ± 0.5
LOS (days)	3.75 ± 1.00
Readmission, *n* (%)	3 (2.9)
Postop wound infection, *n* (%)	7 (6.7)
Postop abscess, *n* (%)	1 (1)

M: male; F: female; BMI: body mass index; WBC: white blood cell count; NEU: neutrophil; CRP: C-reactive protein; BPM: beats per minute, CT: computed tomography, LOS: length of hospital stay.

**Table 2 jcm-13-05504-t002:** The demographic and clinical data of the patients by gender.

Variables	Female (Mean ± SD)	Male (Mean ± SD)	*p*-Value
Age (month)	144.6 ± 48.4	138.1 ± 50.2	0.508
BMI (kg/m^2^)	19.7 ± 4.3	21.4 ± 4.4	0.054
WBC (per µL)	15.6 ± 5.8	14.9 ± 5.0	0.511
NEU (per µL)	12.2 ± 5.5	12.2 ± 5.2	0.970
CRP (mg/L)	77.8 ± 93.2	61.1 ± 64.7	0.283
Symptom—operation time (h)	36.8 ± 37.1	29.8 ± 29.5	0.293
CT–operation time (h)	9.9 ± 12.9	9.2 ± 7.9	0.740
Operation time (min)	50.8 ± 26.7	51.2 ± 16.9	0.929
Appendix diameter (mm)	10.3 ± 2.9	11.5 ± 3.2	0.066
BPM	89.6 ± 8.1	91.0 ± 10.8	0.466
Temperature (°C)	37.60 ± 0.46	37.61 ± 0.48	0.942
LOS (days)	3.88 ± 1.07	3.66 ± 0.92	0.247

BMI: body mass index; WBC: white blood cell; NEU: neutrophil; CRP: C-reactive protein; CT: computed tomography; SD: standard deviation, BPM: beats per minute, LOS: length of hospital stay.

**Table 3 jcm-13-05504-t003:** Correlation analysis of operation time according to continuous variables.

Variables	Correlation Coefficient (*r*)	*p* Value
Age	−0.095	0.336
Height	−0.121	0.222
Weight	−0.034	0.731
BMI	0.007	0.945
WBC	0.106	0.284
NEU	0.240	0.014
CRP	0.295	0.002
Symptom–operation time	0.282	0.004
CT–operation time	0.135	0.173
Appendix diameter	0.234	0.017
BPM	0.031	0.758
Temperature	0.041	0.681
LOS	0.125	0.207
Total score	0.415	<0.001

BMI: body mass index; WBC: white blood cell; NEU: neutrophil; CRP: C-reactive protein; CT: computed tomography, BPM: beats per minute, LOS: length of hospital stay.

**Table 4 jcm-13-05504-t004:** Effect of component removal on model performance for operation time.

Component Removal	R^2^	Adj. R^2^	SEE	F-Value	*p*-Value
All components included	0.359	0.327	17.739	11.001	<0.001
Abscess removed	0.271	0.241	18.883	9.186	<0.001
Extraluminal air removed	0.224	0.200	19.334	9.601	<0.001
Periappendicular fluid removed	0.083	0.065	20.907	4.575	0.013
Periappendicular fat-stranding removed	0.029	0.019	21.413	3.012	0.086

R^2^: coefficient of determination, Adj. R^2^: adjusted R^2^, SEE: standard error of the estimate.

**Table 5 jcm-13-05504-t005:** Results of the univariate and multivariable analysis of the association between the operation time and categorical variables.

Variables	Operation Time (min) (Mean ± SD)	Univariate Analysis *p*-Value	Multivariable Analysis Coefficient (95% CI)	*p*-Value
Appendicolith				
Absent (*n* = 49)	47.2 ± 18.5	0.165	7.30 (−1.04 to 15.64)	0.086
Present (*n* = 55)	54.5 ± 23.7			
Periappendiceal fat standing				
Absent (*n* = 18)	38.9 ± 13.3	0.007	14.77 (3.98–25.56)	0.008
Present (*n* = 86)	53.7 ± 22.2			
Periappendiceal fluid				
Absent (*n* = 62)	43.2 ± 10.1	<0.001	19.71 (12.01–27.40)	<0.001
Present (*n* = 42)	62.9 ± 28.0			
Extraluminal air				
Absent (*n* = 83)	46.3 ± 14.2	<0.001	23.97 (14.55–33.38)	<0.001
Present (*n* = 21)	70.2 ± 33.2			
Abscess		< 0.001		
Absent (*n* = 89)	46.2 ± 13.7		33.76 (23.72–43.79)	<0.001
Present (*n* = 15)	80.0 ± 34.9			
Appendix localization				
Location 1 (*n* = 39)	48.1 ± 16.7			
Location 2 (*n* = 26)	48.7 ± 22.0	0.235	0.57 (−10.23 to 11.38)	0.916
Location 3 (*n* = 39)	55.8 ± 25.2			

Localization of the appendix was classified using three locations: Location 1: just below the anterior abdominal wall; Location 2: in the retrocecal or retro-ascending colon; Location 3: in the pelvis, on the ventral side covered by the small intestine. (R^2^: 0.359).

**Table 6 jcm-13-05504-t006:** Results of the univariate and multivariable analysis of the association between the operation time and continuous variables.

Variables	Value (Mean ± SD)	Univariate Analysis *p*-Value	Multivariable Analysis Coefficient (95% CI)	*p*-Value
Age (month)	140.9 ± 49.3	0.211	−0.02 (−0.117 to 0.065)	0.574
BMI (kg/m^2^)	20.7 ± 4.4	0.945	0.31 (−0.66 to 1.28)	0.524
WBC (per µL)	15.2 ± 5.4	0.317	0.12 (−0.59 to 0.83)	0.736
NEU (per µL)	12.2 ± 5.3	0.029	0.51 (−0.21 to 1.24)	0.162
CRP (mg/L)	68.3 ± 78.4	<0.001	0.05 (0.00–0.11)	0.034
Symptom–operation time (h)	32.8 ± 33.0	<0.001	0.25 (0.10–0.40)	0.001
CT–operation time (h)	9.5 ± 10.3	0.946	−0.46 (−0.90 to −0.02)	0.039
Appendix diameter (mm)	11.0 ± 3.0	0.090	1.42 (0.12–2.72)	0.032

BMI: body mass index; WBC: white blood cell; Neu: neutrophil; CRP: C-reactive protein; CT: computed tomography.

**Table 7 jcm-13-05504-t007:** Results of the multiple regression analysis.

Variables	Coefficient	SD	*t*-Value	CI [0.025–0.975]	*p*-Value
Age	0.03	0.04	0.704	[−0.05 to 0.12]	0.483
Appendix diameter	0.08	0.77	0.108	[−1.45 to 1.62]	0.915
BMI	0.28	0.45	0.623	[−0.61 to 1.18]	0.535
CRP	−0.10	0.02	−0.369	[−0.06 to 0.04]	0.713
NEU/WBC	3.64	20.94	0.174	[−37.95 to 45.24]	0.862
Symptom duration (<2/>2 days)	10.74	0.001	2.351	[0.00 to 0.005]	0.047
CT–operation time	−0.007	0.003	−2.093	[−0.01 to −0.001]	0.039
Appendicolith	0.82	4.16	0.199	[−7.43 to 9.09]	0.843
Periappendiceal fat stranding	6.67	5.20	1.284	[−2.10 to 18.63]	0.203
Periappendiceal fluid	8.54	5.00	1.707	[−1.40 to 18.48]	0.091
Extraluminal air	0.54	7.82	−0.070	[−15.00 to 16.10]	0.944
Abscess	25.31	8.74	2.896	[7.95 to 42.68]	0.005
Localization	3.43	2.11	1.623	[−0.76 to 7.64]	0.108

BMI: body mass index; WBC: white blood cell; Neu: neutrophil; CRP: C-reactive protein; CT: computed tomography. R^2^: 0.433.

**Table 8 jcm-13-05504-t008:** Comparison of clinical and laboratory characteristics between short and long operation time groups.

Variable	Total (*n* = 104) (Mean ± SD/%)	Short Operation Time Group (*n* = 75)	Long Operation Time Group (*n* = 29)	*p*-Value
Operation time (min)	51.11 ± 21.62	41.53 ± 8.65	75.86 ± 25.32	NA
Sex (F/M)	45/59	32/43	13/16	0.842
Age (month)	140.9 ± 49.3	142.2 ± 50.0	137.7 ± 48.2	0.599
Appendix diameter (mm)	11.1 ± 3.0	10.7 ± 3.2	11.9 ± 2.4	0.106
BMI (kg/m^2^)	20.7 ± 4.4	20.6 ± 4.4	21.0 ± 4.6	0.704
CRP (mg/L)	68.4 ± 78.4	54.1 ± 63.9	105.3 ± 99.2	0.002
NEU/WBC	0.73 ± 0.10	0.72 ± 0.10	0.76 ± 0.08	0.118
Symptom–operation time (h)	32.8 ± 33.0	26.9 ± 30.0	48.2 ± 35.0	<0.001
Symptom duration < 2 days	81 (77.9%)	63 (84.0%)	18 (62.1%)	0.016
Symptom duration > 2 days	23 (22.1%)	12 (16.0%)	11 (37.9%)	
CT–operation time (h)	9.5 ± 10.0	9.0 ± 10.0	10.6 ± 9.0	0.059
Appendicolith	55 (52.9%)	37 (49.3%)	18 (62.1%)	0.243
Periappendiceal fat stranding	86 (82.7%)	59 (78.7%)	27 (93.1%)	0.081
Periappendiceal fluid	42 (40.4%)	21 (28.0%)	21 (72.4%)	<0.001
Extraluminal air	21 (20.2%)	8 (10.7%)	13 (44.8%)	<0.001
Abscess	15 (14.4%)	3 (4.0%)	12 (41.4%)	<0.001
Wound infection	7 (6.7%)	3 (4.0%)	4 (13.8%)	0.074
Postop abscess	1 (1%)	0 (0%)	1 (3.4%)	0.106
BPM	90.4 ± 9.7	90.8 ± 9.1	89.4 ± 11.4	0.442
Temperature (°C)	37.61 ± 0.5	37.62 ± 0.5	37.56 ± 0.5	0.497
LOS (days)	3.75 ± 1.0	3.68 ± 0.9	3.96 ± 1.0	0.773
Readmission	3 (2.9%)	2 (2.7%)	1 (3.4%)	0.831

F: female, M: male, BMI: body mass index, CRP: C-reactive protein, WBC: white blood cell; Neu: neutrophil; BPM: beats per minute, LOS: length of hospital stay, SD: standard deviation, NA: not applicable.

**Table 9 jcm-13-05504-t009:** Distribution and statistical comparison of variables based on short and long operation times in non-perforated appendicitis.

Variable		Total (*n* = 82)	Short Operation Time (*n* = 67)	Long Operation Time (*n* = 15)	*p*-Value
Age (month)		150.9 ± 44.2	149.2 ± 46.7	158.9 ± 30.5	0.444
Sex	Female	36 (43.9%)	30 (44.8%)	6 (40%)	0.736
	Male	46 (56.1%)	37 (55.2%)	9 (60%)	
BMI (kg/m^2^)		21.1 ± 4.2	20.8 ± 4.3	22.6 ± 3.3	0.143
CRP (mg/L)		50.2 ± 58.6	48.4 ± 61.2	58.0 ± 46.3	0.570
NEU/WBC		0.72 ± 0.10	0.71 ± 0.10	0.75 ± 0.08	0.217
BPM		90.9 ± 10.0	91.2 ± 9.2	89.9 ± 13.4	0.663
Temperature (°C)		37.6 ± 0.5	37.6 ± 0.5	37.5 ± 0.5	0.730
Symptom–operation time (h)		27.8 ± 27.8	25.9 ± 28.7	36.4 ± 22.4	0.185
Symptom duration	<2 days	68 (82.9%)	57 (85.1%)	11 (73.3%)	0.275
	>2 days	14(17.1%)	10 (14.9%)	4 (26.7%)	
CT–operation time (h)		9.0 ± 10.2	9.0 ± 11.1	9.1 ± 4.6	0.974
LOS (days)		3.75 ± 1.00	3.68 ± 0.95	4.06 ± 1.16	0.185
Appendix diameter (mm)		10.9 ± 3.2	10.7 ± 3.3	11.8 ± 2.6	0.195
Apendicolith	Absent	43 (52.4%)	35 (52.2%)	8 (53.3%)	0.939
	Present	39 (47.6%)	32 (47.8%)	7 (46.7%)	
Periappendiceal fat standing	Absent	17 (20.7%)	15 (22.4%)	2 (13.3%)	0.434
	Present	65 (79.3%)	52 (77.6%)	13 (86.7%)	
Periappendiceal fluid	Absent	62 (75.6%)	54 (80.6%)	8 (53.3%)	0.026
	Present	20 (24.4%)	13 (19.4%)	7 (46.7%)	
Abscess	Absent	82 (100%)	67 (100%)	15 (100%)	NA
	Present	0 (0%)	0 (0%)	0 (0%)	

BMI: body mass index; CRP: C-reactive protein; LOS: length of hospital stay; BPM: beats per minute; NA: not applicable.

**Table 10 jcm-13-05504-t010:** Distribution and statistical comparison of variables based on short and long operation times in perforated appendicitis.

Variable		Total (*n* = 22)	Short Operation Time (*n* = 8)	Long Operation Time (*n* = 14)	*p*-Value
Age (month)		103.5 ± 50.2	83.4 ± 37.4	115.0 ± 54.1	0.161
Sex	Female	9	2 (25%)	7 (50%)	0.251
	Male	13	6 (75%)	7 (50%)	
BMI (kg/m^2^)		19.2 ± 5.0	18.8 ± 4.7	19.4 ± 5.4	0.809
CRP (mg/L)		136.1 ± 103.8	101.2 ± 70.8	156.0 ± 116.3	0.243
NEU/WBC		0.78 ± 0.07	0.79 ± 0.06	0.77 ± 0.08	0.600
BPM		88.4 ± 8.4	87.7 ± 7.7	88.8 ± 9.1	0.789
Temperature (°C)		35.6 ± 0.42	37.6 ± 0.5	37.54 ± 0.4	0.631
Symptom–operation time (h)		51.6 ± 43.6	35.5 ± 41.4	60.8 ± 43.6	0.198
Symptom duration	<2 days	13 (59.1%)	6 (75%)	7 (50%)	0.251
	>2 days	9 (40.9%)	2 (25%)	7 (50%)	
CT–operation time (h)		11.1 ± 11	9.30 ± 9.50	12.25 ± 11.98	0.559
LOS (days)		3.77 ± 0.97	3.62 ± 1.06	3.85 ± 0.94	0.602
Appendix diameter (mm)		11.7 ± 2.2	11.4 ± 2.0	11.9 ± 2.4	0.661
Apendicolith	Absent	6 (27.3%)	3 (37.5%)	3 (21.4%)	0.416
	Present	16 (72.7%)	5 (62.5%)	11 (78.6%)	
Periappendiceal fat standing	Absent	1 (4.5%)	1 (12.5%)	0 (0%)	0.176
	Present	21 (95.5%)	7 (87.5%)	14 (100%)	
Periappendiceal fluid	Absent	0 (0%)	0 (0%)	0 (0%)	NA
	Present	22 (100%)	8 (100%)	14 (100%)	
Abscess	Absent	7 (31.8%)	5 (62.5%)	2 (14.3%)	0.020
	Present	15 (68.2%)	3 (37.5%)	12 (85.7%)	

BMI: body mass index; CRP: C-reactive protein, LOS: length of hospital stay; BPM: beats per minute; NA: not applicable.

## Data Availability

The raw data supporting the conclusions of this article will be made available by the authors upon request.

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
