# Peer review of "Are Preoperative CT Findings Useful in Predicting the Duration of Laparoscopic Appendectomy in Pediatric Patients? A Single Center Study"

_jcm, 2024, doi:10.3390/jcm13185504_

Round 1

Reviewer 1 Report

Comments and Suggestions for Authors

This is a retrospective study aiming to establish the impact of specific CT findings on operation time for appendicitis in children. The authors have made a great effort in re-examining all CT scans and collecting patient data on all these 108 cases. However, I have some issues with the statistical methodology and the conclusions.

It is stated that multivariate linear regression was used although it seems that multivariable regression was used.

It is stated that linear regression was used yet results are presented as odds ratios, how can this be explained?

What is meant by “univariate analysis”? Is it a univariable linear regression or t-test or Mann-Whitney U?

I would have assumed the relationship between time from first symptom to operation and operation time would not be linear instead the first 1-2days would have similar operation time while after around three days it would start getting longer due to abscess formation and the like. Was the assumption of linear relationship tested in any way?

Height, weight and BMI are not independent variables and should not be used in the same multivariable analysis. This should have been noticed if the model was tested for multicollinarity.

The authors states that “the presence of certain clinical features such as periappendiceal fat stranding, periappendiceal fluid, extraluminal air, and abscesses greatly increases surgical time”. However, in the multivariable analysis (table 6) the only CT finding that had a significant impact on operation time was abscess. To me it does not seem accurate to conclude that “using a 5-point scale based on CT findings can accurately anticipate surgical complexity” when only one variable in this 5 point scale influences operation time in the multivariable analysis.

Patient selection for this study was not clear to me. Was all patients in this hospital that had an appendicitis on CT from November 2021 to February 2024 included or was it just patients from a single surgeon? Since appendicitis abscesses are most often treated with drainage and antibiotics, how large a proportion of patients with appendicitis on CT was operated? How large a proportion of patients operated for appendicitis had a preoperative CT? How could patient selection have influenced results?

Author Response

Comment 1: It is stated that multivariate linear regression was used although it seems that multivariable regression was used.

Response 1: Thank you for your valuable comments. We have carefully reviewed your feedback regarding the statistical analysis methods used in our study. The correct term in our manuscript is "multivariable linear regression," but it was incorrectly stated as "multivariate regression." This was purely a writing mistake, and the actual method used was "multivariable regression" analysis.

In our study, we performed a multivariable regression analysis to examine the effects of various independent variables on operation time.

We have revised our manuscript to correct this error to avoid any potential misunderstandings. Thank you again for your thorough review and attention to this matter.

Comment 2: It is stated that linear regression was used yet results are presented as odds ratios, how can this be explained?

Response 2: We have carefully reviewed your feedback regarding the tables presented in our study and identified a editing error. In the tables where the results of the linear regression analysis are presented, the coefficients (B) of the independent variables along with their 95% Confidence Intervals (CI) were provided. However, due to a writing error, the term "OR" (Odds Ratio) was mistakenly used alongside these coefficients.

In fact, the values presented in the tables are "Unstandardized Coefficients (B)" along with their confidence intervals. This error was purely technical and no “Odds Ratios” were calculated in our study. We have reviewed the tables and corrected this error to ensure the coefficients are properly labeled with the correct terminology.

We would like to inform you that we have updated our manuscript to reflect the correct terminology and resolved the misunderstanding. Thank you once again for your valuable feedback.

Comment 3: What is meant by “univariate analysis”? Is it a univariable linear regression or t-test or Mann-Whitney U?

Response 3: We have reviewed our analysis methods and re-examined the accuracy of the applications in our univariate analysis model. First, we would like to clarify that in our study, we used the “simple linear regression” method to evaluate the relationship between the dependent variable (operation time) and each independent variable.

This method was employed to assess the impact of each continuous independent variable on the dependent variable individually. Our aim was to model the effects of the independent variables on operative time as accurately as possible, and we interpreted our results accordingly.

In light of your observations, we would like to confirm that we correctly utilized the simple linear regression method within our univariate analysis model, and we hope these clarifications help to further explain our process.

Thank you once again for your valuable feedback. Your contributions are greatly appreciated and help us improve our work.

Comment 4: I would have assumed the relationship between time from first symptom to operation and operation time would not be linear instead the first 1-2days would have similar operation time while after around three days it would start getting longer due to abscess formation and the like. Was the assumption of linear relationship tested in any way?

Response 4: In response to your valuable feedback, we conducted a series of additional analyses to further explore the relationship between symptom duration and operative time. First, we compared the fit between linear and quadratic regression models to assess their alignment. Additionally, we used the LOESS method to examine nonlinear relationships and created a plot to observe potential clustering within the dataset more clearly. As you pointed out, we found no significant change in operative time during the early days of symptom duration, but a noticeable increase in operative time as the symptom duration extended, particularly after the third day.

Based on these analyses, we revisited our study and categorized symptom duration into two distinct groups: 2 days or less, and more than 2 days, to more clearly define the relationship between symptom duration and operation time. We have revised the results and discussion sections accordingly. Once again, we thank you for your valuable feedback, which has been instrumental in improving our work.

Comment 5: Height, weight and BMI are not independent variables and should not be used in the same multivariable analysis. This should have been noticed if the model was tested for multicollinarity.

Response 5: We have carefully reviewed your comment regarding the use of height, weight, and BMI as independent variables in the same model and have made the necessary adjustments. Upon reassessing our model for multicollinearity, we recognized that including variables such as height, weight, and BMI, which are highly correlated with each other, could negatively impact the results. Therefore, we revised our model by retaining only the BMI variable, which helped to eliminate multicollinearity and improve the accuracy of the model.

Similarly, instead of including both WBC and NEU as separate variables, we used the NEU/WBC ratio, allowing us to better control the relationships between the independent variables and reduce the risk of multicollinearity.

Following these changes, we reran the model and reviewed the results. We have revised the corresponding table to reflect the new analysis and found the results to be more reliable. We are pleased to report that these revisions have made our model more accurate and robust. Our manuscript has been updated accordingly, and we would like to thank you once again for your valuable feedback, which greatly contributed to these improvements.

Comment 6: The authors states that “the presence of certain clinical features such as periappendiceal fat stranding, periappendiceal fluid, extraluminal air, and abscesses greatly increases surgical time”. However, in the multivariable analysis (table 6) the only CT finding that had a significant impact on operation time was abscess. To me it does not seem accurate to conclude that “using a 5-point scale based on CT findings can accurately anticipate surgical complexity” when only one variable in this 5 point scale influences operation time in the multivariable analysis.

Response 6: Thank you for your valuable feedback. You correctly pointed out that, in the results of our multiple regression analysis presented in Table 6, only the presence of an abscess had a significant impact on operation time. This finding highlights that each component of our 5-point CT-based scoring system does not independently influence operative time.

In the analysis to evaluate the effects of removing specific components from the model on operation time, the results showed that the inclusion or exclusion of certain variables significantly impacted model performance. When all components were included, the model showed an R² of 0.359, indicating that 35.9% of the variance in operation time was explained by the predictors, with an F-value of 11.001 (p < 0.001). Removing the abscess variable resulted in a drop in R² to 0.271 and an increased standard error of the estimate (SEE) to 18.883. Further removals of extraluminal air and periappendicular fluid continued to decrease the model’s explanatory power, with the removal of periappendicular fluid lowering R² to 0.083 and the F-value to 4.575 (p = 0.013). Finally, removing periappendicular fat-stranding reduced R² to 0.029, with the model’s performance becoming marginally significant (F = 3.012, p = 0.086). Based on the analysis, the abscess variable is the most important factor in the model, and its removal significantly decreases the model’s explanatory power and performance (Table 4).

In light of these findings, we would like to revise our manuscript to clarify the limitations of our scoring system in predicting operative time. “The 5-point CT scoring system used in our study can serve as a guide for predicting overall surgical complexity, but each component's independent effect on operative time is limited. Specifically, only the presence of an abscess was significantly associated with prolonged operative time. Therefore, we have revised the discussion section to acknowledge the limitations of our scoring system in predicting operative time more explicitly.”

Once again, thank you for your valuable contribution, which has been instrumental in improving our manuscript.

Comment 7: Patient selection for this study was not clear to me. Was all patients in this hospital that had an appendicitis on CT from November 2021 to February 2024 included or was it just patients from a single surgeon? Since appendicitis abscesses are most often treated with drainage and antibiotics, how large a proportion of patients with appendicitis on CT was operated? How large a proportion of patients operated for appendicitis had a preoperative CT? How could patient selection have influenced results?

Response 7: Thank you for your valuable comments. To clarify the patient selection process in our study, we provide the following details:

Our study included 104 patients who were diagnosed with appendicitis via contrast-enhanced CT and underwent laparoscopic appendectomy at our hospital between November 2021 and February 2024. All surgeries were performed by the same pediatric surgeon at our hospital, thereby eliminating the potential variability in operation time due to different surgeons.

During this period, a total of 138 patients were diagnosed with appendicitis via CT at our hospital. Surgical intervention was performed on 127 of these patients, while the remaining patients were treated with non-surgical methods, such as drainage or antibiotic therapy.

Out of 386 total appendectomy patients, 127 underwent preoperative CT evaluation. The remaining patients were evaluated using ultrasound imaging. Eight patients with non-contrast CT scans, nine with incomplete imaging records, and four with missing medical records were excluded from the study. Additionally, two patients with severe comorbidities that could have influenced operative time were also excluded.

As our study is retrospective, patient selection has certain limitations. However, since all surgeries were performed by a single surgeon, variability in operation time due to the operator was minimized. We believe that the criteria used in patient selection ensured a homogeneous distribution of surgical complexity. Additionally, the exclusion of patients treated with drainage or antibiotics did not affect the overall conclusions regarding operation time, as these patients followed a different treatment pathway.

We have revised our manuscript accordingly and provided a more detailed explanation of the patient selection process in materials and methods section. Thank you once again for your valuable contributions.

Reviewer 2 Report

Comments and Suggestions for Authors

This manuscript by Ismail Taskent, et al. investigated the correlation between preoperative CT findings and the duration of laparoscopic appendectomy (LA) in pediatric patients. The authors retrospectively reviewed the medical records of 104 pediatric patients diagnosed with acute appendicitis via contrast-enhanced CT who subsequently underwent laparoscopic appendectomy (LA) between November 2021 and February 2024. The authors conclude that specific CT findings, such as periappendiceal fat stranding, fluid, air, and abscesses, are strongly associated with prolonged operation times, with abscesses having the most significant impact.

The study question is valid and relevant in clinical practice. The research design is appropriate, and the study methods are adequately described. The authors of the manuscript have done a good job of formulating the study plan and clearly presenting their findings. However, the manuscript is missing several important data that could have influenced the outcomes. Please consider the following suggestions for the improvement of this manuscript.

- Define prolonged operation time for LA in children in this study. Include data for normal and prolonged operation times for various appendectomy procedures, including complicated cases like perforated appendicitis.

- Include normal values for CRP in your lab.

- Include other clinical variables that could have influenced outcomes like duration of symptoms, presence of sepsis signs like fever/tachycardia, need for blood transfusion at OR, presence of comorbidities, etc.

-  How many patients had perforated appendicitis and what factors predicted prolonged operation times in these cases.

- Study the difference in postoperative outcomes for normal vs prolonged operation times in your cohort.

Author Response

Comment 1: Define prolonged operation time for LA in children in this study. Include data for normal and prolonged operation times for various appendectomy procedures, including complicated cases like perforated appendicitis.

Response 1: Based on the distribution analysis we conducted, the median operation time was determined to be 50 minutes. Operation times exceeding 50 minutes were classified as prolonged. Using this threshold, we performed various analyses to identify the factors influencing longer and shorter operation times. The results showed that both clinical factors and CT findings were significantly associated with prolonged and shorter operation times, suggesting that these factors could be used to predict surgical duration.

In response to your feedback, we have revised our study and provided a more detailed explanation of our analyses (Table 8). Thank you for your valuable insights.

Comment 2: Include normal values for CRP in your lab.

Response 2: In our study, C-reactive protein (CRP) levels were included using the normal reference range of “0-5 mg/L” as defined by our laboratory. We appreciate your contributions and thank you for your valuable feedback.

Comment 3: Include other clinical variables that could have influenced outcomes like duration of symptoms, presence of sepsis signs like fever/tachycardia, need for blood transfusion at OR, presence of comorbidities, etc.

Response 3: Thank you for your valuable comments. In our study, important clinical variables such as symptom duration were carefully considered and analyzed. Additionally, clinical parameters related to sepsis, such as fever and heart rate, were included in our analysis. We confirm that none of our patients required a blood transfusion, and we will specify in the methods section that patients with comorbidities, which could influence the results, were excluded from the study. Once again, we sincerely appreciate your valuable contributions.

Comment 4: How many patients had perforated appendicitis and what factors predicted prolonged operation times in these cases.

Response 4: In our study, 22 patients were diagnosed with perforated appendicitis. In the analyses conducted by separating the perforated and non-perforated groups, it was found that the most significant factor predicting operative time in perforated appendicitis cases was the presence of an abscess on CT, while in the non-perforated group, the most important factor was the presence of periappendiceal fluid. We have updated our study accordingly and presented our results in more detail (Table9,10). Thank you once again for your valuable comments and feedback.

Comment 5: Study the difference in postoperative outcomes for normal vs prolonged operation times in your cohort.

Response 5: Based on your suggestions, we divided our patients into early and late groups according to operative times and analyzed the clinical and laboratory findings, including postoperative outcomes. We have incorporated these results into our study and revised our manuscript according to the new findings (Table 8). Thank you once again for your valuable comments and feedback.

Round 2

Reviewer 1 Report

Comments and Suggestions for Authors

In my opinion the authors have made a great effort in revising the manuscript including redoing many of the analysis. The use of statistics and statistical terms is much more solid. Even though the manuscript is longer now I still find it easier to read and more focused and convincing. The term multivariate is still used (probably by mistake) in one place in the discussion but apart from that I do not have anything to add at this point.

Reviewer 2 Report

Comments and Suggestions for Authors

The authors addressed my concerns incorporated my suggestions.